# Sparse Annotations with Random Walks for U-Net Segmentation of Biodegradable Bone Implants in Synchrotron Microtomograms

**Niclas Bockelmann**[1],                                         NICLAS.BOCKELMANN@STUDENT.UNI-LUEBECK.DE
**Diana Krüger**[2], **D.C. Florian Wieland**[2], **Berit Zeller-Plumhoff**[2], **Niccoló Peruzzi**[4], **Silvia Galli**[5], **Regine Willumeit-Römer**[2,3], **Fabian Wilde**[2], **Felix Beckmann**[2], **Jörg Hammel**[2], **Julian Moosmann**[2],                  JULIAN.MOOSMANN@HZG.DE
**Mattias P. Heinrich**[1]                                        HEINRICH@IMI.UNI-LUEBECK.DE

[1] *Institute for Medical Informatics, University of Luebeck*

[2] *Institute of Materials Research, Helmholtz-Zentrum Geesthacht*

[3] *Institute of Materials Science, Christian-Albrechts-University Kiel*

[4] *Clinical Sciences, Lund University*

[5] *Faculty of Odontology, Malmö University*

## Abstract

Currently, most bone implants used in orthopedics and traumatology are non-degradable and may need to be surgically removed later on e.g. in the case of children. This removal is associated with health risks which could be minimized by using biodegradable implants. Therefore, research on magnesium-based implants is ongoing, which can be objectively quantified through synchrotron radiation microtomography and subsequent image analysis. In order to evaluate the suitability of these materials, e.g. their stability over time, accurate pixelwise segmentations of these high-resolution scans are necessary. The fully-convolutional U-Net architecture achieves a Dice coefficient of $0.750 \pm 0.102$ when trained with a small dataset with dense expert annotations. However, extending the learning to larger databases would require prohibitive annotation efforts. Hence, in this work we implemented and compared new training methods that require only a small fraction of manually annotated pixels. While directly training on these scribble annotation deteriorates the segmentation quality by 26.8 percentage points, our new random walk-based semi-automatic target achieves the same Dice overlap as a dense supervision, and thus offers a more promising approach for sparse annotations.

## 1. Introduction

Titanium and its alloys are most commonly used for permanent bone implants in traumatology and orthopedics due to their mechanical properties and good biocompatibility (Moosmann et al., 2017). However, to reduce surgical risks and complications in case of later implant removal for growing patients (e.g. children), biodegradable implants - magnesium-based screws - are studied in animal experiments and imaged using synchrotron radiation microtomography to gain a better understanding of the ossointegration and degradation (Moosmann et al., 2017) and this analysis relies on accurate automated segmentation of 3D volumes. Deep learning, more specifically has become the method of choice in medical

image analysis, but usually requires large and densely labeled datasets (Litjens et al., 2017). Pixel-wise segmentations are most commonly obtained using the U-Net (Ronneberger et al., 2015) or V-Net (Milletari et al., 2016) architectures. The required dense label annotation of 3D training images is a time consuming and expensive process for which expert knowledge is necessary. For that reason a training setup with scribble-supervision has been proposed by Can et al. (Can et al., 2018) where a random walk algorithm is employed in conjunction with a recurrent neural network and a conditional random field. In this paper, we demonstrate that (for our given task) a simplified pipeline for sparse annotation learning with high accuracy can be designed using the following steps: 1) a multi-slice 2.5D U-Net (cf. (Mehta and Sivaswamy, 2017)) is employed that considers neighbouring slices of the input volume for a single annotated segmentation slice 2) an appropriately tuned random-walk algorithm propagates sparse scribble based labels to a dense image grid, and 3) a morphological processing step that removes local label noise. In contrast to (Can et al., 2018) these steps already match the dense annotation quality and make the time-consuming iterative training process of the recurrent networks obsolete.

## 2. Material and Methods

**Data:** The data consists of four data sets in which a screw has been implanted into a bone. These microtomogram data sets were acquired with high energy synchrotron radiation and contain at least 250 axial slices (further referenced as z dimension) of which only 25 slices have been labeled (every 10th slice). Four labels are used: "background", "bone", "corroded screw" and "screw".

**Network Architecture:** The used network for pixel-wise segmentation is oriented on the multi-slice variant of the 3D U-Net architecture of (Ronneberger et al., 2015) proposed by (Mehta and Sivaswamy, 2017). The network is modified in the way that a 3D input volume with eight slices in the $z$ dimension is cropped out of the full volume: four slices below and above the corresponding data slice are extracted. Therefore only partially labelled volumes can be used for our training. Each 3D convolution throughout the network is followed by a Group Normalization layer with a group number of two and a leaky rectified linear unit. By processing the volume through the network the $z$ dimension is reduced to a size of 1, which enables a supervision with individual 2D segmentations.

**Training:** The weighted cross-entropy loss between the multi-channel output and the label image is used as supervision. All training was done with data augmentation (including affine transformations), Adam optimization with a learning rate of 0.005, a duration of 250 training epochs and a batch size of two.

**Experiments:** We conducted a leave-one-out cross-validation over all four available 3D data sets. To evaluate the performance the Dice coefficient is obtained and as an optional preprocessing step a binary closing on the label "bone" for the target image of the network is evaluated. Two experiments each, one with and one without the usage of preprocessing are conducted for each supervision method. We evaluate dense annotations, where the complete target label is provided. Subsequently, we explore scribbled label segmentations for which dense annotations are automatically obtained using a random walk algorithm (Grady, 2006) using the grayvalue scan for edge-preservation. As a last approach the network is trained directly with the scribble annotations by ignoring the loss of non-scribbled voxels.

Table 1: Averaged cross validation results in Dice.

| Training Target | Preprocessing | Total | Bone | Corroded Screw | Screw |
|---|---|---|---|---|---|
| Dense annotation | No | $0.750 \pm 0.102$ | $0.825 \pm 0.062$ | $0.538 \pm 0.137$ | $0.888 \pm 0.093$ |
| Dense annotation | Yes | $0.703 \pm 0.109$ | $0.756 \pm 0.078$ | $0.472 \pm 0.137$ | $0.880 \pm 0.103$ |
| Random walk | No | $0.687 \pm 0.106$ | $0.743 \pm 0.102$ | $0.415 \pm 0.127$ | $0.905 \pm 0.086$ |
| **Random walk** | Yes | $\mathbf{0.751} \pm 0.068$ | $0.798 \pm 0.063$ | $0.541 \pm 0.095$ | $0.916 \pm 0.029$ |
| Scribble | No | $0.482 \pm 0.106$ | $0.568 \pm 0.098$ | $0.293 \pm 0.126$ | $0.585 \pm 0.088$ |

## 3. Results and Discussion

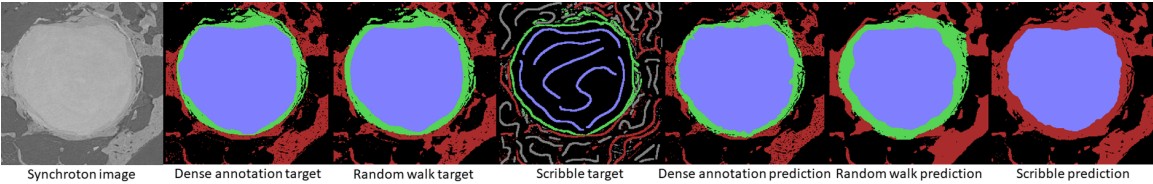

Synchroton image    Dense annotation target    Random walk target    Scribble target    Dense annotation prediction   Random walk prediction    Scribble prediction

Figure 1: Axial view of screw implant in bone. Black = background, red = bone, green = corroded screw, blue = screw.

The quantitative results of the experiments with respect to the different training targets are provided in Table 1. Supervising the network training with a densely annotated image followed by a morphological closing step does not yield advantages over not using preprocessing. However, when using the sparsely annotated labels and a random walk generated target image for training, this preprocessing step is crucial to achieve highly accurate predictions. Comparing our proposed sparsely supervised method with the best settings for a U-Net trained with dense annotations, only a small difference in Dice of 0.1 percentage points is observed. The approach using only scribbled labels as targets for training showed a degradation in Dice of 26.8 percentage points with respect to training the U-Net with dense annotation images. It can be seen that the Dice coefficient for the "Corroded Screw" is substantially lower thatn that of the other labels. This remaining challenge is caused by the relatively small area of "Corroded Screw" and the difficult differentiation of surrounding regions in terms of gray value. A qualitative visualisation of the results is given in Fig. 1.

## 4. Conclusion

In this paper, different methods for learning a dense U-Net prediction from sparse label annotations is evaluated for the task of segmenting microtomograms of screw bone implants, which were acquired using synchroton radiation. Using a random-walk algorithm for an edge-preserving propagation of scribble labels to a whole image slice was found to be substantially better than directly training on sparse annotations. In addition, a morphological pre-processing step was shown to yield further large improvements and achieved the overall best results with a Dice of $0.751 \pm 0.068$, which matches the quality of a densely supervised U-Net. A remaining challenge is the corrosion area, which may require more variable training data to perform better.

## Acknowledgments

Data was acquired by BMBF projects SynchroLoad (project number 05K16CGA) and Mg-Bone (project number 05K16CGB) which are funded by the Rntgen-ngstrm Cluster (RC), a bilateral research collaboration of the Swedish government and the German Federal Ministry of Education and Research (BMBF). We acknowledge provision of beamtime at beamline P05 at PETRA III at DESY, a member of the Helmholtz Association (HGF)

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
