# OpenReview forum: "Sparse Annotations with Random Walks for U-Net Segmentation of Biodegradable Bone Implants in Synchrotron Microtomograms"
_MIDL.io/2019/Conference/Abstract — MIDL Abstract 2019_

### Official Review · AnonReviewer1 · 2019-04-28
**effective solution to weakly (scribble-based) supervision - improve on evaluation (other metrics than Dice)**

**Rating:** 3
**Confidence:** 3

**Review:**

Sparse annotations with random walks for u-net segmentation of biodegradable bone implants in synchrotron microtomograms

The paper proposes to exploit partial annotations, in the form of scribbles, to segment bone implant structures in synchrotron images. The method is inspired by a weakly supervised method of Can, DLMIA'18, where random walks bootstraps a learning approach. Comparative results indicates the effectiveness of the method, where scribble-based training sets are up to par with fully supervised approaches.
The results could perhaps be better highlighted by using a complementary evaluation metric. Dice scores, indeed, don't reveal much on the quality of the boundaries.

---

### Official Review · AnonReviewer2 · 2019-04-29
**Well-written abstract, but methodological contribution and validation seem limited**

**Rating:** 2
**Confidence:** 2

**Review:**

The authors show that training a U-net for segmentation with random walk generated labels from scribbles works as well as using fully annotated datasets. Although I think this is an interesting message, I think the methodological contribution of the abstract is not strong enough. The random walk method to go from scribbles to labels was previously developed and here just applied. The resultant label maps look like fully annotated cases so the fact that this works well is not surprising. The validation is also limited due to the availability of just 4 cases.

---

### Decision · Program_Chairs · 2019-05-06
**Acceptance Decision**

Accept